# Biosynthesis of Silver Nanoparticles by *Conyza canadensis* and Their Antifungal Activity against *Bipolaris maydis*

Yueming Yi [1,†], Changjin Wang [1,†], Xinxin Cheng [1], Kechuan Yi [2], Weidong Huang [1,*] and Haibing Yu [1,*]

[1] College of Agriculture, Anhui Science and Technology University, Chuzhou 233100, China; Yiym@ahstu.edu.cn (Y.Y.); wangcj@ahstu.edu.cn (C.W.); chengxx@ahstu.edu.cn (X.C.)

[2] College of Mechanical Engineering, Anhui Science and Technology University, Chuzhou 233100, China; Yikc@ahstu.edu.cn

\* Correspondence: huangwd@ahstu.edu.cn (W.H.); yuhb@ahstu.edu.cn (H.Y.)

† These authors contribute equally.

**Abstract:** Silver nanoparticles were biosynthesized from *Conyza canadensis* leaf extract with the help of a microwave oven. The UV-vis spectrum showed the maximum absorption at 441 nm, corresponding to the surface plasmon resonance of silver nanoparticles. Transmission electron microscope and scanning electron microscope images showed that the synthesized silver nanoparticles were spherical or near-spherical with an average diameter of 43.9 nm. X-ray diffraction demonstrated nanoparticles with a single-phase cubic structure. As-synthesized silver nanoparticles displayed prominent antifungal activity against *Bipolaris maydis*. The colony inhibition rate reached 88.6% when the concentration of nanosilver colloid was 100 $\mu L \cdot mL^{-1}$ (*v/v*). At such a concentration, no colony formation was observed on the solid plate. The diameter of the inhibition zone was $13.20 \pm 1.12$ mm. These results lay the foundation for the comprehensive control of plant pathogens using an environmentally friendly approach.

**Keywords:** antimicrobial activity; silver nanoparticles; *Conyza canadensis*; *Bipolaris maydis*



## 1. Introduction

The use of nanotechnology, which appeared later than bio-engineering, has expanded in the last few decades [1–3]. It makes traditional substances into nanoscale materials, which display excellent performance, including optical, magnetic, physical, and chemical characteristics, compared with their bulk counterparts [4–6]. Among these materials, silver nanoparticles have proved to be one of the most important types. Consequently, significant research has been conducted on the synthesis, characterization, and application of silver nanoparticles [7–14].

Silver was used to disinfect and dress wounds in ancient times; its antimicrobial property has drawn much attention since then. Silver nanoparticles improve the antimicrobial activity of silver and have proven to be efficient. They inhibit bacteria, fungi, and viruses. Silver nanoparticles synthesized using the chemical approach exhibited an excellent antibacterial effect against *Staphylococcus aureus* (ATCC 6538p) at 50 $\mu g \cdot mL^{-1}$; the bacteria cells, enzymatic activity, and proteins changed [15]. The minimum bactericidal concentration of silver nanoparticles synthesized by *Descurainia sophia* against *Agrobacterium tumefaciens* (strain GV3850) and *A. rhizogenes* (strain 15843) was 4 and 8 $\mu g \cdot mL^{-1}$, respectively [16]. The silver nanoparticles synthesized by *Phoenix dactylifera* showed a significant inhibitory effect on *Rhizoctonia solani* (AG2_2) cultures; the concentration of 25 $\mu g/mL$ prevented approximately 83% of the mycelium growth of the fungus [17]. Silver nanoparticles synthesized using turnip leaf extract interacted with several wood-degrading fungal pathogens, such as *Gloeophyllum abietinum*, *Gloeophyllum trabeum*, *Chaetomium globosum*, and *Phanerochaete sordida*. The result showed cellulose and lignocellulose degradation for all of the tested fungi [18]. The in vitro anti-HBV (Hepatitis B Virus) activities of silver nanoparticles (about

10 and 50 nm) were determined using the HepAD38 cell line. The results showed that silver nanoparticles could inhibit the in vitro production of HBV RNA and extracellular virions [19]. The anti-HIV (Human Immunodeficiency Virus) activity of silver nanoparticles was proven in the early stage of viral replication and post-entry stages of its life cycle; the results made it possible to inhibit a wide variety of circulating HIV-1 strains [20].

In this study, the silver nanoparticles biosynthesized from *Conyza canadensis* extract were applied to determine their antifungal activity against *Bipolaris maydis*. Fungicides have made an undeniable contribution to disease control. However, in the current scenario, several drawbacks became apparent, such as pesticide residue, environmental pollution, pathogens resistance, and so forth. It is necessary to explore neotype and stable fungistats to replace or assist fungicides. As non-antibiotic agents, silver nanoparticles could overcome such problems to the greatest degree.

## 2. Materials and Methods

### 2.1. Fungus Isolation and Plant Tissue

The single-spore isolate of *B. maydis* was preserved in the Laboratory of Plant Protection at Anhui Science and Technology University. *C. Canadensis* was gathered from the plantation of the same campus.

### 2.2. Biosynthesis of Silver Nanoparticles

Air-dried *C. Canadensis* leaves were cut into tiny pieces (1 cm × 1 cm) after washing them thoroughly with distilled water. About 10 g of leaves was added to 100 mL of deionized water and heated in a microwave oven at 800 W for 10 min. $AgNO_3$ solution (100 mmol·$L^{-1}$) was freshly prepared as stock solution. The leaf extract was filtered with Whatman No.1 paper. After that, 1 mL leaf filtrate and 1 mL $AgNO_3$ stock solution were added to 98 mL deionized water, making the final concentration of $AgNO_3$ 1 mmol·$L^{-1}$. Finally, the mixture was put into a microwave oven and heated at 800 W for 2 min. The color and UV-vis spectrum of the solution were measured.

### 2.3. Characterization of Silver Nanoparticles

Several analysis tools, such as a UV-vis absorption spectrometer, transmission electron microscope (TEM), scanning electron microscope (SEM), and X-ray powder diffractometer (XRD), and energy-dispersive X-ray spectroscopy (EDX) were applied to determine the maximum absorption peak, morphology, crystal structure, particle size, and size distribution so as to obtain the detailed information of synthesized silver nanoparticles.

### 2.4. Antifungal Activity of Silver Nanoparticles against B. maydis

2.4.1. Inhibition of Colony Growth

Different volumes (1, 2, 3, 4, and 5 mL) of nanosilver colloid and varied volumes (4, 3, 2, 1, and 0 mL) of sterile water were added to 45 mL melted PDA (Potato dextrose agar) separately, and the concentration of nanosilver colloid was set as 20, 40, 60, 80, and 100 μL·$mL^{-1}$ (*v/v*), respectively. The control contained 45 mL melted PDA medium and 5 mL sterile water. Fungus blocks (φ = 5 mm) were drilled from cultivated *B. maydis*, and one specimen was placed in the center of each Petri dish, then inoculated, followed by incubation at 28 °C for 3–5 days. Each control and treatment was performed in three replicates. The inhibition rate of silver nanoparticles against *B. maydis* was calculated using the following equation.

$$\text{Inhibition rate} = [(\text{Diameter of control colony} - \text{Diameter of treatment colony})/(\text{Diameter of control colony} - \text{Diameter of fungus block})] \times 100\%$$

2.4.2. Influence on Colony Formation

The PDA plates contained different concentrations (20, 40, 60, 80, and 100 μL·$mL^{-1}$, *v/v*) of nanosilver colloids, respectively. About 45 mL melted PDA medium with 5 mL sterile water was set as control. The spores' suspension of the *B. maydis* isolate was adjusted to $10^6$ $mL^{-1}$ with a counting chamber. Then, 100 μL of the spore suspension was spread

evenly on cooled PDA plates using a sterile spreading rod. Each control and treatment was performed in three replicates. Finally, the plates were incubated at 28 °C for 2–3 days after standing for 3–5 min.

### 2.4.3. Measurement of the Inhibition Zone

The inhibition zone created near the agar well that was drilled into the PDA plate displayed the antifungal activity of silver nanoparticles. About 100 µL of the spore suspension ($10^6$ mL$^{-1}$) was evenly smeared on a solid PDA plate. Five wells ($\varphi$ = 8 mm) were drilled by a hole puncher, and different volumes (20, 30, 40, and 50 µL) of nanosilver colloid and 50 µL sterile water were dripped into corresponding agar well. After incubation for several days, an inhibition zone appeared near the agar well if the filling liquid had antifungal activity. The diameter of the inhibition zone varied as well. However, no inhibition zone was created near the well that contained sterile water. Each control and treatment was performed in three replicates. The plates were incubated at 28 °C for 2–3 days after standing for 3–5 min.

## 3. Results and Discussion

### 3.1. Biosynthesis of Silver Nanoparticles

Silver ($Ag^0$) was formed from $AgNO_3$ ($Ag^+$) and plant extract. Figure 1 shows that the solution color changed from faint yellow to reddish-brown after adding 1 mmol·L$^{-1}$ $AgNO_3$, indicating the formation of silver nanoparticles [21,22]. The maximum absorption peak appeared at 441 nm, according to the surface plasmon resonance of silver nanoparticles [18]. Silver nanoparticles could be synthesized by physical, chemical, and biological methods. Among these methods, the last one is preferred by researchers owing to its unique virtues, such as the abundance of synthesis materials, easy operation, eco-friendly nature, and so forth [23–27]. Hence, the biosynthesis of silver nanoparticles by the microwave-assisted method has been the least time-consuming approach until now.

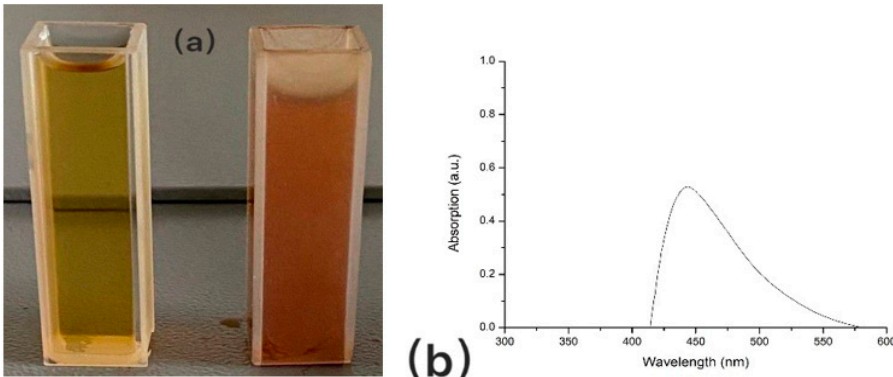

**Figure 1.** Biosynthesis of silver nanoparticles mediated by *C. Canadensis*. (**a**) Image of leaf extract without and with $AgNO_3$, (**b**) UV-vis absorption spectrum of silver nanoparticles.

### 3.2. TEM Analysis of Silver Nanoparticles

Figure 2a–c shows that silver nanoparticles synthesized from the *C. canadensis* leaf extract were primarily near-spherical, with rare aggregations. For definite particle size and size distribution, 200 particles were randomly selected for measurement using Image J software. The results showed that the biosynthesized silver nanoparticles were between 18.1 and 75.0 nm, with an average size of 43.9 nm (Figure 2d).

### 3.3. SEM Analysis of Silver Nanoparticles

The whole morphology of silver nanoparticles was scanned using SEM. The images are shown in Figure 3a–c. The EDX graph in Figure 3d demonstrates the existence of silver, while the other signals, such as K, C, and O, may be attributed to the *C. canadensis* leaf extract on the substrate.

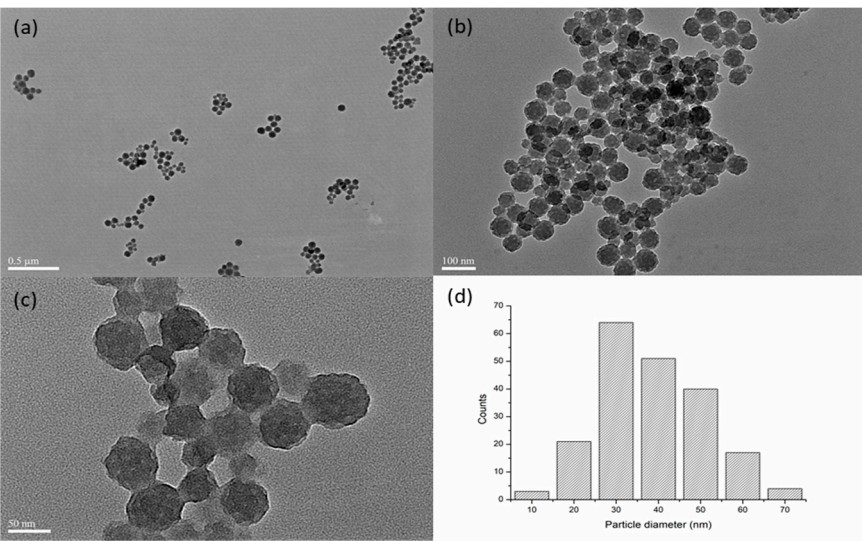

**Figure 2.** TEM images (**a**–**c**) and size distribution (**d**) of silver nanoparticles.

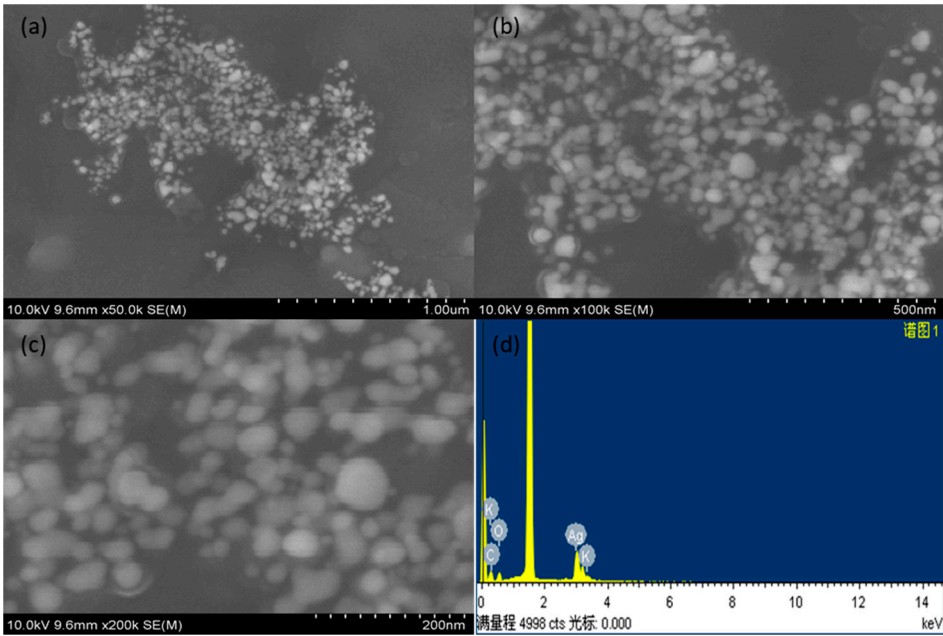

**Figure 3.** SEM images (**a**–**c**) and EDX spectrum (**d**) of silver nanoparticles.

### 3.4. XRD Analysis of Silver Nanoparticles

The XRD pattern of biosynthesized silver nanoparticles is shown in Figure 4. It demonstrates the existence of Ag with a monoclinic crystalline system. The reflections with 2θ values of 22.26°, 26.52°, 28.18°, 30.88°, 32.62°, 36.80°, 43.10°, 44.80°, and 63.20° observed on the spectrum should be indexed to silver faces of (111), (200), and (220) [28]; the 2θ value of 76.10° might be related to (420) plane [29].

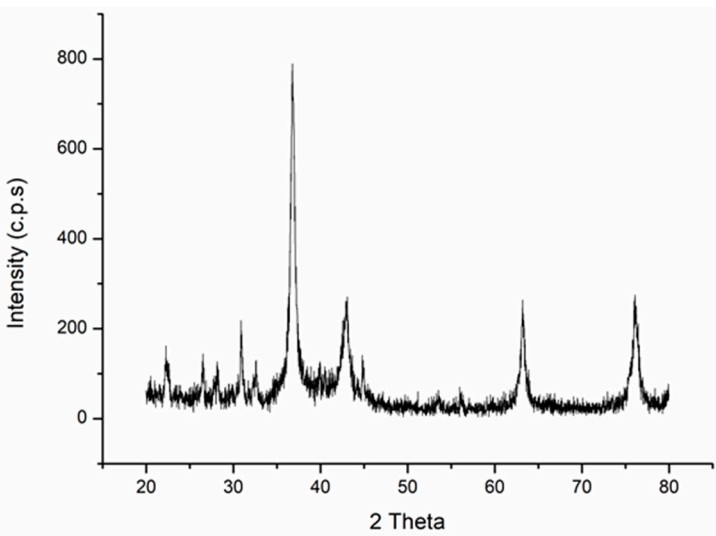

**Figure 4.** XRD pattern of silver nanoparticles.

### 3.5. Inhibition of Colony Growth

Figure 5 shows that various concentrations of silver nanoparticles exhibited prominent antifungal activity against the colonies of *B. maydis*. For the control, the colony diameter was 7.5 cm, and it decreased dramatically as the concentration of silver nanoparticles increased. When the concentration of silver nanoparticles was 100 $\mu L \cdot mL^{-1}$, the diameter reached its minimum value of 1.3 cm, and the inhibition rate reached 88.6%.

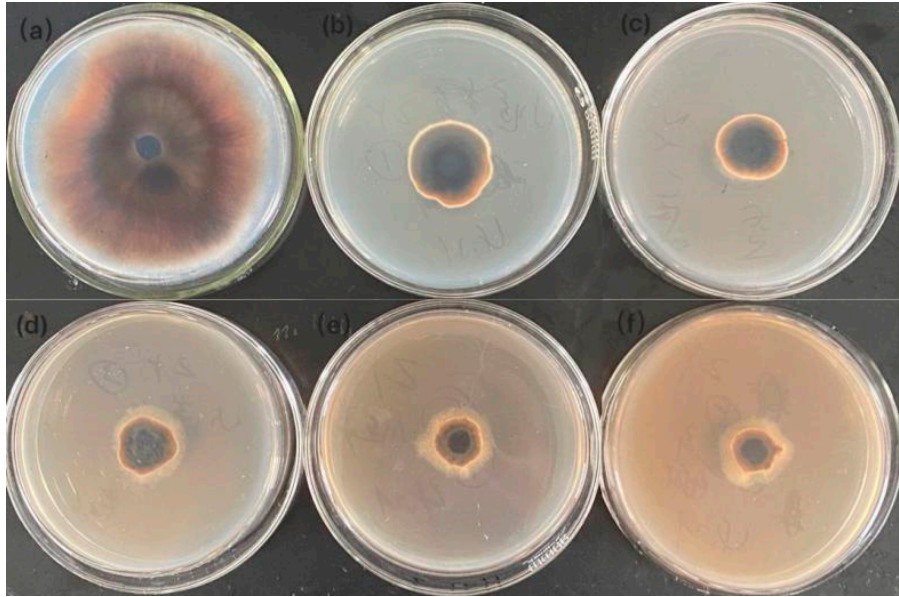

**Figure 5.** Inhibition of the colony growth of silver nanoparticles against *B. maydis*. (**a**) control isolate, (**b**–**f**) various concentration (20, 40, 60, 80, and 100 $\mu L \cdot mL^{-1}$, *v/v*) of silver nanoparticles.

### 3.6. Influence on Colony Formation

Figure 6 shows that that colony of *B. maydis* was dramatically inhibited by silver nanoparticles. A mass of colonies appeared all over the control plate. When the concentration of silver nanoparticles increased from 20 to 100 $\mu L \cdot mL^{-1}$ (*v/v*), no colony formation occurred for all the treatments. The excellent antimicrobial activity of silver nanoparticles, including antibacterial, antifungal, and antiviral activities, has been widely demonstrated [19,20,30–34]. In fact, not only do the same silver nanoparticles show a diverse inhibitory effect against various pathogens [35–37], but silver nanoparticles synthesized by different methods exhibited different inhibitory activity against the same pathogen [16,17].

Although several studies reported the antifungal effect of silver nanoparticles against phytopathogens, the effect against *B. maydis* has rarely been reported. The inhibition effect of silver nanoparticles could be influenced by several parameters, such as the concentration of AgNO$_3$, pH, quantities of synthesis materials, stabilizers, modified substances, and so forth.

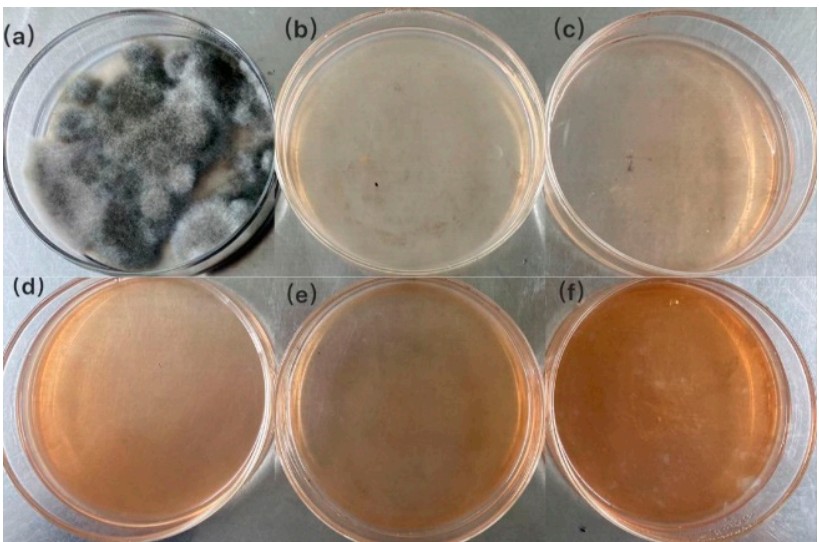

**Figure 6.** Colony formation of *B. maydis* on PDA plates. (**a**–**f**) different concentration (0, 20, 40, 60, 80, and 100 μL·mL$^{-1}$, *v/v*) of silver nanoparticles.

### 3.7. Inhibition Zone Diameter

The diameter of the inhibition zone also reflected the antimicrobial activity against fungus or bacteria. As shown in Table 1, the colony and mycelia of *B. maydis* covered the agar well without silver nanoparticles, with no inhibition zone. An obvious inhibition zone was created near the agar wells that contained different volumes of silver nanoparticles; it reached 13.20 ± 1.12 mm when the well was filled with 50 μL of silver nanoparticles. Besides the agar well diffusion approach used to measure the diameter of the inhibition zone, the paper disk diffusion method was also applied; the antimicrobial activity varied based on different nanomaterials, pathogens, and so forth [38,39].

**Table 1.** Diameter of inhibition zone of silver nanoparticles against *B. maydis*.

| Volume of Silver Nanoparticles (μL) | Inhibition Zone Diameter (mm) |
|:---:|:---:|
| 0 | 0.00 ± 0.00 |
| 20 | 7.00 ± 0.87 |
| 30 | 8.60 ± 2.05 |
| 40 | 11.20 ± 1.86 |
| 50 | 13.20 ± 1.12 |

### 4. Conclusions

In this study, silver nanoparticles were biosynthesized from the *C. Canadensis* leaf extract using the microwave approach. Several analysis tools, such as UV-vis, TEM, SEM, XRD, and EDX, were applied to characterize synthesized silver nanoparticles systematically. The antifungal activity of silver nanoparticles against *B. maydis* was tested through colony growth, colony formation, and inhibition zone. All the results demonstrated that the synthesized silver nanoparticles exhibited prominent antifungal activity against *B. maydis*. As is known to us, *C. Canadensis* plays two main roles: one is as a farmland weed that does great harm to agriculture, and the other is as Chinese herbal medicine that has an anti-inflammatory effect. This study developed the value of *C. Canadensis*, turning waste into treasure. The results also provide a novel approach and fungistat to control plant

pathogens, and it contributes to reducing the dosage of chemical pesticides to alleviate agriculture problems, such as environmental pollution and drug residue.

**Author Contributions:** Conceptualization, W.H. and H.Y.; methodology, Y.Y. and X.C.; formal analysis, C.W. and K.Y.; investigation, Y.Y. and C.W.; writing—original draft preparation, Y.Y. and C.W.; writing—review and editing, W.H. and H.Y.; project administration, X.C. and K.Y. All authors have read and agreed to the published version of the manuscript.

**Funding:** This work was funded by the Key Research and Development Program of Anhui Province (202004a06020004), Natural Science Fund of Education Department of Anhui Province (KJ2018A0543, KJ2019A0814, KJ2019ZD57), Talent Introduction Project in Anhui Science and Technology University (NXYJ201602).

**Institutional Review Board Statement:** Not applicable.

**Informed Consent Statement:** Not applicable.

**Data Availability Statement:** Available on request.

**Conflicts of Interest:** The authors declare no conflict of interest.

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
