# Peer review of "Biosynthesis of Silver Nanoparticles by Conyza canadensis and Their Antifungal Activity against Bipolaris maydis"

_crystals, doi:10.3390/cryst11121443_

Round 1

Reviewer 1 Report

Dear Author. I read the manuscript. The paper well writing and have good quality of presentation. I strongly suggest for publication after some correction. Some major concern should be answered.

Title: Why you did not use antifungal? Antimicrobial is not scientific for this research.

In introduction: "is known that silver was used to disinfect and dress wound in ancient time, and the antimicrobial property has been drawing much attention from then on" Please add reference for this claim.

"Figure 1. (b) UV-vis absorption spectrum of silver nanoparticles" is not normal. Why the peack started from 410 nm? Normally did not like this.

TEM is vary good.. Can you add TEM with lower scale bar such as 20 nm?

In introduction please add short paragraph about importance of plants and bio-Engineering using this references: doi: 10.1016/j.matdes.2021.109683 , 37(9), 1319-1321. doi: 10.1093/bioinformatics/btaa832 , doi: 10.1186/s40538-021-00214-x , https://doi.org/10.1186/s40538-021-00255-2

Figure 5. Why control isolate did not cover all over of plate? Can you add picture that cover all culture media?

Author Response

Dear Reviewer,

Thank you very much for giving me a chance to revise my manuscript (crystals-1452512). According to the comments of the reviewer, I try my best to modify it, please forgive me if there were inappropriate or incomplete places.

Reviewer 2 Report

The present manuscript describes the synthesis of silver nanoparticles in water with the aid of an extract of Conyza Canadensis and heating in the microwave. The formed nanoparticles are rather large and are shown to inhibit grown of the fungus Bipolaris maydis in seemingly high concentrations. The results seem to indicate that the spores of this fungus are destroyed. The manuscript itself is interesting, but premature. There are many technical errors that should be corrected prior to publication.  

  1. Please let this manuscript revise by a native speaker. The English is not acceptable for publication.
  2. The idea of synthesizing silver nanoparticles using environmentally friendly conditions should be discussed in more detail to point out the new aspects of this study. Please also mention a previous work, where silver nanoparticles were synthesized in polylysine using ascorbic acid as reducing agent (Ho et al. Journal of Biomaterials Science, Polymer Edition 24 (13), 1589-1600 (2013)).
  3. The experimental part is unclear. The synthesis is not described in detail. For example, the silver concentration in the extract is not clear. Since silver particles are solid, there will never be a concentration given as 10% v/v. Please provide the concentration as µg silver/mL or as mol silver/L. Further, the DNA and protein release conditions should be described in more detail. Does the fungus grow under these conditions in the suspension, is there a kinetics in the release experiment or are the results simply an increased absorbance due to the plant extract? The kinetics of the release should be provided.
  4. Is there any optimization or development in design? How is the concentration of silver or plant extract influencing the size and the yield of the silver nanoparticles?
  5. Figure 1: What are the 2 cuvettes? Why is the spectrum not complete?
  6. Figure 3: What is the Y-axe of the EDX-spectrum. Please translate the chinese characters there.
  7. The authors mention in the conclusion that their work will help against formation of resistant strains. Since it is known that silver nanoparticles induce resistance, this statement should be removed.

Author Response

(The authors gave the same response as above.)

Round 2

Reviewer 2 Report

The manuscript has been improved by the authors and their comments are sufficient with the important exception that the experimental part is still unclear. Therefore the manuscript is not scientifically sound in its present form.

  1. The silver concentration is essential to judge the results. To my understanding, the authors could simply calculate it. The experimental part says that “followed by the addition of 1 mL of the filtrate and 1 mmol.L-1 69 AgNO3” I have no idea, what that means. How much extract was mixed with how much silver nitrate solution in which concentration? If you provide this information, the silver concentration in the final solution and the diluted solution can be calculated. This concentration must be given in the manuscript and not the useless 10% v/v of something unknown.  
  2. How were the agar plates prepared? How much of the silver nanoparticle stock solution was added to how much growth medium/agar mixture? Please provide details that would allow other researcher to repeat your experiments.
  3. The inhibition zone test was performed as follows: “About 100 μL of the spore suspension was smeared evenly on a solid Five wells represented different volumes (20, 30, 40, and 50 μL) of silver nanoparticles.Sterile water waspoured on the plate, followed by the dripping of corresponding nanoparticles in each well.”  To me this means that the spores are evenly distributed on the plate. If you add an un-known amount of sterile water and different volumes of an un-known silver nanoparticle solution, there is no information of the nanoparticle concentration. Please provide an experimental description that even possibly allows a repetition of the experiments. What are the wells? Are those agar plates, which diameter, is the liquid distributed over the whole plate?
  4. The protein and DNA release is simply measured by UV absorbance. There is no control, no kinetics, no prove that this is not simply the increasing concentration of the plant extract, because the concentration of this is increased with that of the nanoparticles. If the authors would measure a UV/Vis of the silver nanoparticle solution with water as reference, they would find out that the plant extract is showing high absorbance in the UV region. Therefore I do not believe that there is any DNA or protein release from the spores, but this is simply an artefact of a non-sound scientific work. The only possibility to prove that there is anything releasing using the method of the authors is to provide a release kinetics. If the authors are not willing to do that they should remove this unsound statement of DNA and protein release from the manuscript.  
  5. There are still many typos in the manuscript.

Author Response

Dear Reviewer,

Thank you very much for reviewing my manuscript (crystals-1452512) again. According to the comments, I try my best to modify it, please forgive me if there were inappropriate or incomplete places.
